# HumBugDB: a large-scale acoustic mosquito dataset

**Ivan Kiskin**[*]
University of Oxford

**Marianne Sinka**[†]
University of Oxford

**Adam D. Cobb**[||]
SRI International

**Waqas Rafique**[*]
University of Oxford

**Lawrence Wang**[*]
University of Oxford

**Davide Zilli**[¶]
Mind Foundry Ltd

**Ben Gutteridge**[§]
University of Oxford

**Rinita Dam**[†]
University of Oxford

**Theodoros Marinos**[††]
University of Surrey

**Yunpeng Li**[††]
University of Surrey

**Dickson Msaky**[‡]
IHI Tanzania

**Emmanuel Kaindoa**[‡]
IHI Tanzania

**Gerard Killeen**[**]
UCC, BEES

**Kathy Willis**[†]
University of Oxford

**Steve Roberts**[*]
University of Oxford

[*]Dept. Eng. Science: {ikiskin, waqas, sjrob}@robots.ox.ac.uk,
lawrence.wang@eng.ox.ac.uk, [†]Dept. Zoology: {marianne.sinka,
kathy.willis,rinita.dam}@zoo.ox.ac.uk, [||]adam.cobb@sri.com,
[††]{tm00591, yunpeng.li}@surrey.ac.uk, [**]gerard.killeen@ucc.ie,
[¶]davide.zilli@mindfoundry.ai, [§]benjamin.gutteridge@new.ox.ac.uk
[‡]Ifakara Health Institute: {dmsaky,ekaindoa}@ihi.or.tz.

## Abstract

This paper presents the first large-scale multi-species dataset of acoustic recordings of mosquitoes tracked continuously in free flight. Mosquitoes are well-known carriers of diseases such as malaria, dengue and yellow fever. The motivation for collecting such a large dataset comes from the need to gather information, help predict outbreaks, and inform data-driven policy. The task of detecting mosquitoes from their wingbeats is made challenging due to the difficulty in collecting recordings from realistic scenarios. To address this, as part of the HumBug project, we have conducted global experiments to record mosquitoes ranging from those bred indoors in culture cages to mosquitoes captured in the wild. As a result, the audio recordings vary widely in signal-to-noise ratio and contain a broad range of indoor and outdoor background environments from Tanzania, Thailand, Kenya, the USA and the UK. The audio recordings have been labelled by domain experts, aided by Bayesian neural networks. As a result, we present 20 hours of mosquito audio recordings expertly labelled with tags precise in time, of which 18 hours are annotated from 36 different species. We provide our data from a regularly maintained database, which captures important metadata such as the capture method, age, feeding status and gender of the mosquitoes. Additionally, we provide code to extract features and train Bayesian convolutional neural networks that can distinguish mosquito sounds from their corresponding background. Our contribution is to provide a dataset that is both challenging to machine learning researchers focusing on acoustic identification, and critical to entomologists, geo-spatial modellers and other domain experts to understand mosquito behaviour, model their distribution, and manage the threat they pose to humans.

Submitted to the 35th Conference on Neural Information Processing Systems (NeurIPS 2021) Track on Datasets and Benchmarks. Do not distribute.

# 1   Introduction

There are over 100 genera of mosquito in the world containing over 3,500 species and they are found on every continent except Antarctica [Harbach, 2013]. Only one genus (*Anopheles*) contains species capable of transmitting the parasites responsible for human malaria. *Anopheles* contain over 475 formally recognised species, of which approximately 75 are vectors of human malaria, and around 40 are considered truly dangerous [Sinka et al., 2012]. These 40 species are inadvertently responsible for more human deaths than any other creature. In 2019, for example, malaria caused around 229 million cases of disease across more than 100 countries resulting in an estimated 409,000 deaths [World Health Organization, 2020]. It is imperative therefore to accurately locate and identify the few dangerous mosquito species amongst the many benign ones to achieve efficient mosquito control. Mosquito surveys are used to establish vector species' composition and abundance, human biting rates and thus the potential to transmit a pathogen. Traditional survey methods, such as human landing catches, which collect mosquitoes as they land on the exposed skin of a collector, can be time consuming, expensive, and are limited in the number of sites they can survey. They can also be subject to collector bias, either due to variability in the skill or experience of the collector, or in their inherent attractiveness to local mosquito fauna. These surveys can also expose collectors to disease. Moreover, once the mosquitoes are collected, the specimens still need to undergo post sampling processing for accurate species identification. Consequently, an affordable automated survey method that detects, identifies and counts mosquitoes could generate unprecedented levels of high-quality occurrence and abundance data over spatial and temporal scales currently difficult to achieve. It is for this reason that we utilise low-cost smartphones as acoustic mosquito sensors to solve this task. The exponential increase in smartphone ownership is a worldwide phenomenon. Governments and independent companies are continuing to extend connectivity across the African continent [Friederici et al., 2017]. More than half of sub-Saharan Africa is expected to be connected to a mobile service by 2025 [GSMA, 2020]. With this expanding coverage of mobile phone networks across Africa, there is an emerging opportunity to collect huge datasets, as exemplified by the World's Bank Listening to Africa Initiative [World Bank Organisation, 2017]. Our target application (Section 3.1) uses a free downloadable app, which means that every smartphone can be a mosquito monitor.

**Our contribution**   In order to assist research in methods utilising the acoustic properties of mosquitoes, as part of the HumBug project (described in Section 3.1) we contribute:

- **Data:** http://doi.org/10.5281/zenodo.4904800: A vast database of 20 hours of finely labelled mosquito sounds, and 15 hours of associated non-mosquito control data, constructed from carefully defined recording paradigms. Data was collected over the course of five years in a global collaboration with mosquito entomologists. Recordings were captured from 36 species (or species complexes[1]) with a mix of low-cost smartphones and professional-grade recording devices, to capture both the most accurate noise-free representation, as well as the sound that is likely to be recorded in areas most in need. A diverse range of wild and lab culture mosquitoes is included to capture the biodiversity of naturally occurring species. Our data is stored and maintained in a PostgreSQL database, ensuring label correctness and data integrity. We export all of the audio across a vast range of experiments with a single line in Python, and the metadata we require for experiments with a single SQL query (Appendix C). This allows us to add to our database and re-release data in a reliable and efficient manner.

- **Code:** https://github.com/HumBug-Mosquito/HumBugDB: Detailed tutorial code for training state-of-the-art baseline Bayesian neural network models (a range of ResNet and deep CNN models) for the task of distinguishing mosquitoes of any species from their background surroundings, such as other insects, speech, urban, and rural noise. This baseline model was used to automatically tag a subset of mosquito recordings in this database with a very low false positive rate, by making use of uncertainty metrics such as the predictive entropy and mutual information [Kiskin et al., 2021].

- To ensure learnt models are tested on diverse and realistic data splits, we withheld two test sets: one which captures free-flying mosquitoes around specifically adapted bednets

---

[1]Species complexes are closely related sibling species that are morphologically identical but can have hugely diverse behaviours that allows one to be a prominent and dangerous vector, and another to be harmless.

(mimicking the intended target application as closely as possible), and another which contains caged mosquitoes recorded in free flight in very challenging noisy conditions.

The rest of the paper is structured as follows. Section 2 details related datasets and describes how ours contributes to the literature uniquely. Section 3 shows the primary intended use case for the data and model released in this paper for our overall aims to assist in the eradication of insect-borne diseases. Section 4 describes in detail the sources and collection methods of data present, as well as how and why we perform our train-test split. Section 5 suggests additional use cases for the data, and details the steps taken to train a benchmark model, including an overview of feature extraction, model training and evaluation code. We discuss the results that our models achieve, and the open challenges remaining that our test sets motivate. We conclude by summarising our contribution to various communities in Section 6.

We provide comprehensive instructions for using our baseline models and feature extraction code in Appendix B, and supply additional details on all the metadata in Appendix C. The datasheet (Appendix D) details the dataset's composition (D.2), the data acquisition process (D.3), preprocessing (D.4), past and suggested use cases (D.5), sources of data bias and mitigation strategies (D.6), and database maintenance policies (D.7).

## 2 Related work

Mosquitoes have particularly short, truncated wings allowing them to flap their wings faster than any other insect of equivalent size – up to 1,000 beats per second [Simões et al., 2016, Bomphrey et al., 2017]. This produces their very distinct flight tone and has led many researchers to try and use their sound to attract, trap or kill them [Perevozkin and Bondarchuk, 2015, Johnson and Ritchie, 2016, Jakhete et al., 2017, Fanioudakis et al., 2018, Mukundarajan et al., 2017]. However, there have been very few large datasets released to the public to aid this research. We summarise key statistics of a range of datasets available publicly in Table 1, and discuss the varying sensor modalities separately due to their inherent differences in acoustic properties.

Table 1: A comparison of related mosquito acoustic and pseudo-acoustic datasets released publicly. The *'Average mosquito length'* is the approximate length of audible mosquito recording per sample. This length can not be estimated for Mukundarajan et al. [2017], as the data is crowdsourced, unlabelled and uncurated. Crowdsourced data recording or labels are marked with (*). *'Type'* format: majority, (minority), represents if the mosquitoes have been captured as individuals in the wild, or grown and reproduced in controlled conditions in lab colonies. Where not known, *'Mosquito'* is estimated from the mosquito average mosquito sample duration multiplied by the number of positive samples in dataset.

| Dataset | Sensor | Mosquito (Background) | Average mosquito length | Species | Type |
|---|---|---|---|---|---|
| Chen et al. [2014, UCR] | Opto-acoustic | 17 min (N/A) | $\approx 0.02$ s | 6 | Lab |
| Fanioudakis et al. [2018] | Opto-acoustic | 39 hr (N/A) | $\approx 0.5$ s | 6 | Lab |
| Vasconcelos et al. [2020] | Acoustic | 15 min (N/A) | 0.3 s | 3 | Lab |
| Mukundarajan et al. [2017] (*) | Acoustic | N/A (N/A) | N/A | 20 | Lab, (wild) |
| Kiskin et al. [2019, 2020] (*) | Acoustic | 2 hr (20 hr) | 1 s | N/A | Lab, (wild) |
| **HumBugDB** | Acoustic | 20 hr (15 hr) | 9.7 s | 36 | Wild, (lab) |

**Opto-acoustic approaches** *'Wingbeats'* [Fanioudakis et al., 2018] and *'UCR Flying Insect Classification'* [Chen et al., 2014] are high-SNR pseudo-acoustic datasets collected via optical sensors. We note this is a different, but complementary, approach. Due to the directionality of the recording

method, typical sample durations are encountered from "only a few hundredths of a second" [Chen et al., 2014] to approximately half a second [Fanioudakis et al., 2018]. The approach therefore does not capture the acoustical properties of mosquito sound in free flight which aid mosquito detection in purely acoustic approaches [Vasconcelos et al., 2020]. Furthermore, these datasets survey lab-grown mosquito colonies which do not capture the biodiversity of mosquitoes encountered in the wild [Huho et al., 2007, Hoffmann and Ross, 2018].

**Acoustic approaches** The authors of a recent acoustic mosquito dataset [Vasconcelos et al., 2020] motivated its release by stating that none of the published datasets include environmental noise, which is essential to fully characterise mosquitoes in real-world scenarios. Their dataset consists of 300 ms snippets, amounting to a total of 15 minutes of mosquito recordings. This is an excellent first step. However, for deep learning algorithms the dataset is not readily useable due to its size. Moreover, state-of-the-art models for acoustic classification use training example sizes of at least 0.96 seconds for a variety of audio event detection tasks [Hershey et al., 2017] and often greater depending on the importance of long-range temporal context [Pons et al., 2017, Pons and Serra, 2019, Shimada et al., 2020]. Our dataset consists of mosquito samples with an average duration of 10 seconds and, additionally, we supply equal quantities of corresponding background to form a balanced class distribution of mosquito and noise (see Section 4).

Mukundarajan et al. [2017] have released an acoustic dataset recorded in free flight with smartphones. However, due to a lack of a rigorous recording protocol, the subsequent quality of the recordings is inconsistent, and there is a lack of metadata recording external factors which influence mosquito sound. There are no labels to exactly timestamp the mosquito events in files where mosquito sound is only sporadic, detracting from the overall utility of the dataset. Our database is specifically designed to eliminate these issues based on previous experience with acoustic mosquito recordings.

Kiskin et al. [2019, 2020] released extensive data spanning 22 hours of audio recordings, with crowdsourced labels covering overlapping two-second sections. However, of these, only 2 hours were labelled as containing mosquito sound. In addition, the accuracy of the labels is unknown, and the task of labelling was made difficult as clips were presented in isolation, lacking the expert knowledge and relevant background information that specialists utilised for their labels. Curated data of that release is a subset of the release of this paper, in which we improve upon the past release thanks to a dedicated joint effort between the zoological and machine learning communities.

Nevertheless, we do stress that experimentation which combines information from all of the datasets found in the literature is highly encouraged, and may help find solutions to cover multiple recording modalities, such as opto-acoustic and smartphone acoustic sensors.

## 3 Data for mosquito-borne disease prevention

### 3.1 The HumBug project

The HumBug project is a collaboration between the University of Oxford, Royal Botanic Gardens, Kew, and mosquito entomologists worldwide [HumBug, 2021]. One of the goals of the project is to develop a mosquito acoustic sensor that can be deployed into the homes of people in malaria-endemic areas to help monitor and identify the mosquito species, allowing targeted and effective vector control. Due to the rarity of mosquito events, as part of the pipeline we require a robust method for distinguishing mosquito events from background noise. This constitutes the primary use case for the baseline models of Section 5. We discuss alternate use cases further in Section 5 and Appendix D.5. In the following paragraphs we describe the role of our overall pipeline of Figure 1 by each component.

**Capturing mosquito with smartphones** We developed a power-efficient app to record mosquito flight tone using the in-built microphone on a smartphone (MozzWear [Marinos et al., 2021]). We used 16-bit mono PCM wave audio sampled at 8,000 Hz, based on prior acoustic low-cost smartphone recording solutions for mosquitoes [Li et al., 2017b, Kiskin et al., 2018].[2] To make mosquitoes fly close enough to a smartphone, we have developed an adapted bednet that utilises the inherent behaviour of host-seeking mosquitoes (Figure 2) [Sinka et al., 2021, Sec. 2.1.2]. The combination of

---

[2]The latest version records in 32 kbps `aac` in Tanzanian rural areas where bandwidth is critically limited.

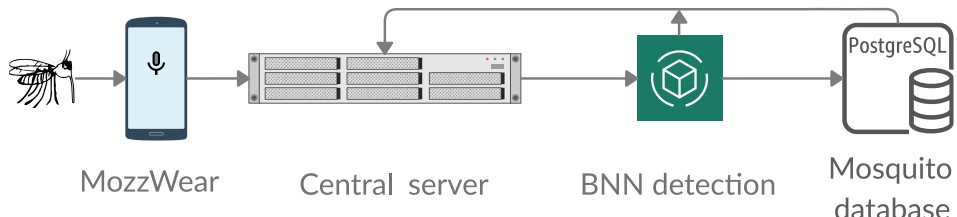

MozzWear     Central server     BNN detection     Mosquito database

Figure 1: Schematic of project workflow. MozzWear is the mobile phone application used to capture the audio. The app synchronises to a central server, where audio enters the BNN model. Successful detections are used to updated a curated database. Information feeds back to improve the model.

the bednets and smartphones constitutes the intended use case, for which we construct Test set A (see Table 2).

**Central server**  Following app recording, audio is synchronised by the app, automatically or initiated by the user, to a central file server for the storage of sound recordings, and a MongoDB [MongoDB Inc, 2021] instance for the storage of metadata. The server possesses a frontend dashboard where recordings and predictions fed back from the model can be accessed. The unstructured nature of the NoSQL engine allows for additional flexibility in storing metadata, especially when new information becomes available.

**BNN detection**  The classification engine deploys a Bayesian convolutional neural network (BCNN), which provides predictions with uncertainty metrics [Kiskin et al., 2021] with Monte Carlo (MC) dropout [Gal and Ghahramani, 2016]. The raw predictions of the model are fed back to the central server, and positive predictions alongside uncertainty estimates are accessible via an HTML dashboard. Positive predictions are then filtered by the probability, mutual information and predictive entropy [Houlsby et al., 2011], screened, and stored in a curated database. This drastically reduces the time spent labelling by domain experts – for our bednet data recorded in Tanzania, we estimate 1 to 2 % of 2,000 hours of recorded data contained mosquito events. Finding these events without assistance from the model was infeasible due to the vast quantity of data.

**PostgreSQL database**  Due to the complex requirements of variables and data storage, we designed a relational database in PostgresSQL [PostgreSQL Global Development Group, 2021], which ensures a standardisation in the labelling and metadata process. The main concept is that all audio is stored on a data server, and each recording is uploaded with a unique ID (the full specifics are included in the database documentation provided in Appendix C). The rigorous structure of this database allows us to validate data input and ensure consistency throughout the schema. This mitigates a major cause of data quality issues and time costs in field studies. Recordings are stored in wave format at their respective sample rates, and all the metadata in csv format. For our maintenance policy, details of ethics agreements, and detailed documentation refer to the datasheet for datasets (Appendix D).

**Privacy**  As a subset of data from the database may contain human speech, and other types of personal data (e.g. data recorded during trials where smartphones were actively listening continuously), we include in this paper only audio which has been assigned an explicit label of *'mosquito', 'audio', 'background'*, or otherwise full consent from members was obtained (for example where entomology experts state a recording ID, and ambient conditions etc.). Additionally, since labels have been generated both by hand and with the use of mosquito detection algorithms, to ensure no speech that has not had explicit consent for release was included in the dataset, we performed voice activity detection using Google's WebRTC project [Ramirez et al., 2007], which is open-source, lightweight, reliable and fast [Ali, 2018, Karrer, 2020]. Sahoo [2020] tested the WebRTC VAD method over 396 hours of data, across multiple recording types. The approach was between 77 % and 99.8 % accurate. Any mosquito labels which overlapped with speech labels were removed, without truncating or re-sampling any audio to keep the format of the data in the database consistent.

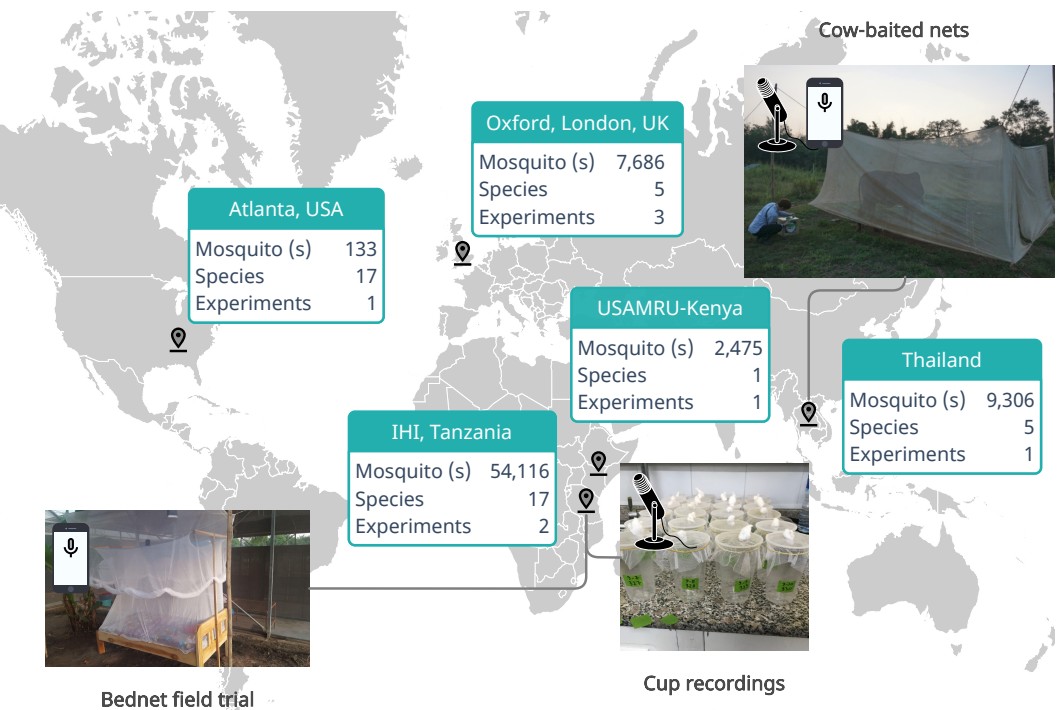

Figure 2: Map of aggregated data acquisition sites.

## 4 The HumBugDB dataset

### 4.1 Summary

Our large-scale multi-species dataset contains recordings of mosquitoes collected from multiple locations globally, as well as via different collection methods. Figure 2 shows the different locations, with the availability of labelled mosquito sound (in seconds) and number of species, and the number of experiments conducted at each location. In total, we present 71,286 seconds (20 hours) of labelled mosquito data with 53,227 seconds (15 hours) of corresponding background noise to aid with the scientific assessment process, recorded at the sites of 8 experiments. Of these, 64,843 seconds contain species metadata, consisting of 36 species (or species complexes) with the distributions illustrated in Appendix C, Figure 6 and Table 6. Table 2 gives a more detailed summary of the type of mosquitoes that were captured, and Appendix C gives a complete explanation of every field in the metadata.

In the following section we break down the data sources according to the nature of mosquitoes – bred within laboratory culture (Section 4.2.1) or wild (Section 4.2.2). We discuss the recording device and the environment the mosquitoes were recorded in – free flying in culture cages, free flying in cups or free flying in bednets (HumBug adapted bednets [Sinka et al., 2021, Sec. 2.1.2]). We also detail the methods of capture (applicable to wild mosquitoes only). These involve traditional mosquito sampling methods, including larval collection, human-baited nets (HBN), adapted Center for Disease Control Light Traps (CDC-LTs) and animal-baited nets (ABN). The method of capture is documented in more detail in Appendix C. We also make clear which dataset is used for training, and which set of experiments is used for testing the models of Section 5.

### 4.2 Data collection

#### 4.2.1 Laboratory culture mosquitoes

Many institutes that conduct research into mosquito-borne diseases hold laboratory cultures of common vector species. These include primary malaria vectors (e.g. *Anopheles gambiae*, *An. arabiensis*), arbovirus vectors including primary vectors of dengue virus (*Aedes albopictus*), yellow fever virus (*Aedes aegypti*) and west nile virus (*Culex quinquefasciatus*). The controlled conditions

Table 2: Key audio metadata and train-test partition. *'Wild'* mosquitoes captured and placed into paper *'cups'* or attracted by bait surrounded by *'bednets'*. *'Culture'* mosquitoes bred specifically for research. Total length (in seconds) of mosquito recordings per group given, with the availability of species meta-information in parentheses. Total length of corresponding non-mosquito recordings, with matching environments, given as *'Negative'*. Full metadata given in Appendix C.

| Data (mosquitoes) | Site (country) | Recorded in | Device (sample rate) | Mosquito (s) (with species) | Negative (s) |
|---|---|---|---|---|---|
| **Train** (wild) | Kasetsart (Thailand) | cup (2018) | Telinga (44.1 kHz) | 9,306 (2,869) | 7,896 |
| **Train** (wild) | IHI (Tanzania) | cup (2020) | Telinga (44.1 kHz) | 45,998 (45,998) | 5,600 |
| **Train** (culture) | Zoology (Oxford, UK) | cup (2017) | Telinga (44.1 kHz) | 6,573 (6,573) | 1,817 |
| **Train** (culture) | LSTMH (UK) | cup (2018) | Telinga (44.1 kHz) | 376 (376) | 147 |
| **Train** (culture) | CDC (USA) | cage (2016) | phone (8 kHz) | 133 (127) | 1,121 |
| **Train** (culture) | USAMRU (Kenya) | cage (2016) | phone (8 kHz) | 2,475 (2,475) | 31,930 |
| **Test A** (culture) | IHI (Tanzania) | bednet (2020) | phone 8 kHz | 4,118 (4,118) | 3,979 |
| **Test B** (culture) | Zoology (Oxford, UK) | cage (2016) | phone (8 kHz) | 737 (737) | 2,307 |
| All | All | All | All | 71,286 (64,843) | 53,227 |

of laboratory cultures produce uniformly sized fully-developed adult mosquitoes which are used for a variety of purposes, including trialling new insecticides or examining the genome of these insects.

**UK, Kenya, USA**  Although the intrinsic variability found amongst natural populations of mosquitoes is not present in laboratory cultures, they do provide access easily to multiple species of concern. Thus we made recordings from the laboratory cultures at the London School of Tropical Medicine and Hygiene (LSTMH), the United States Army Medical Research Unit-Kenya (USAMRU-K), the Center for Diseases Control and Prevention (CDC), Atlanta, as well as with mosquitoes raised from eggs in our own laboratories at the Department of Zoology, University of Oxford. These primary recordings allowed us to quickly evaluate whether flight tone could allow us to distinguish between different species [Li et al., 2018]. Mosquitoes were recorded by placing a recording device into the culture cages where one or multiple mosquitoes were flying, or by placing individual mosquitoes into large cups and holding these close to the recording devices.

We reserve one set of these recordings taken in culture cages by Zoology, Oxford, as one of our test datasets (denoted Test B in Table 2), as past models were able to achieve excellent mosquito detection performance when trained on data held out from the same experiment [Kiskin et al., 2018, 2017]. In this paper we treat this experiment as disparate from the remaining data, increasing the difficulty of the detection task considerably.

**Tanzania**  To fulfill the aim of targeted vector control through the deployment in people's homes, we need to be able to passively capture the mosquito's flight tone. Therefore, in our database we include mosquitoes passively recorded in the Ifakara Health Institute's semi-field facility (*'Mosquito City'*) at Kining'ina, that most closely resembles the intended use of the HumBug system. It is for this reason that a labelled subset (by an expert zoologist with the help of positive BCNN predictions) of this data forms our primary test set, also marked as Test A in Table 2.

The facility houses six chambers containing purpose-built experimental huts, built using traditional methods and representing local housing constructions, with grass roofs, open eaves and brick walls. Four different configurations of the HumBug Net [Sinka et al., 2021], each with a volunteer sleeping

under the net, were set up in four chambers. Budget smartphones were placed in each of the four corners of the HumBug Net (Figure 2). Each night of the study, 200 laboratory cultured *An. arabiensis* were released into each of the four huts and the MozzWear app began recording.

### 4.2.2 Wild captured mosquitoes

Wild mosquitoes naturally exhibit far greater intra-specific variability. To study how this affects our ability to distinguish different species, we conducted experiments in Thailand and Tanzania.

**Thailand**    Across the malaria endemic world, Asia has more dominant vector species (mosquitoes whose abundance or propensity to bite humans makes them particularly efficient vectors of disease) and species complexes anywhere else. Mosquitoes were sampled using ABNs (cow-baited nets in Figure 2), HBNs and larval collections over a period of two months during peak mosquito season (May to October 2018). Sampling was conducted in Pu Teuy Village at a vector monitoring station owned by the Kasetsart University, Bangkok. The mosquito fauna at this site include a number of dominant vector species, including *An. dirus* and *An. minimus* alongside their siblings (*An. baimaii* and *An. harrisoni*) respectively (Appendix C, Figure 6 and Table 6 show the exact species distribution). Mosquitoes were collected at night, carefully placed into large sample cups and recorded the following day using the high-spec Telinga field microphone and a budget smartphone (Appendix D.3 for device details).

**Tanzania**    While Asia has the most diverse vector community, sub-Saharan Africa has the most dangerous and efficient mosquito species, namely *An. gambiae*. This is the species often referred to as the 'most dangerous animal in the world' and as a consequence, sub-Saharan Africa has the highest transmission of human malaria in the world, and the highest number of deaths [World Health Organization, 2020]. Using the methodology trialled in Thailand and with the help of our collaborators at the Ifakara Health Institute, we began a collection and recording project in the Kilombero Valley, Tanzania. HBNs, larval collections and CDC-LTs were used to sample wild mosquitoes and record them in sample cups in the laboratory. *An. gambiae* and *An. funestus* (another highly dangerous mosquito found across sub-Saharan Africa), are also siblings within their respective species complexes. Thus, standard polymerase chain reaction (PCR) identification techniques [Scott et al., 1993] were used to fully identify mosquitoes from these groups.[3] For all the cup recordings in Thailand and Tanzania, environmental conditions (temperature, humidity) were monitored throughout the recording process. The Tanzanian sampling has collected 17 different species including: *An. arabiensis* (a member of the *gambiae* complex), *An. coluzzii*, *An. funestus*, *An. pharoensis* (see Appendix C, Figure 6, Table 6 for a full breakdown).

## 5    Benchmark

To showcase the utility of the data, we supply baseline models that function as acoustic mosquito event detectors. Other use cases include, but are not limited to, species classification, harmonic analysis, and the study of inter-species variability. For a more thorough consideration of these use cases refer to Appendix D.5. We discuss possible data biases arising from species imbalance, mosquito types, and multiple recording devices, and suggest mitigation strategies in Appendix D.6. For the task of mosquito event detection, we hold out Test set A of labelled field data which most closely resembles the target application. Achieving good performance on that set does not guarantee good scalability to other use cases in itself, and for this reason we use Test set B – a shorter, but very difficult low-SNR dataset as a performance marker. The prominent species in this experiment is also not as well represented, providing a further challenge. The statistics of the training and test sets are given in the rows of Table 2. In the upcoming section we will give an overview of the code we supply for our benchmarks. In Section 5.2 we describe the steps taken to train our models, and in Section 5.3 we detail how we define the performance metrics and evaluate the models supplied.

### 5.1    Code use

The top-level Jupyter notebook (Appendix B for data directory tree, code access, and layout) performs data partitioning, feature extraction and segmentation in `get_train_test_from_df()`, model

---

[3]The database gives the PCR identification within the `species` column, or the genus/complex if not available.

training in `train_model()`, and model evaluation in `get_results()`. The code is configured with `config.py`, where data directories are specified for the data, metadata and outputs, and feature transformation parameters are supplied. Model hyperparameters are given in `config_keras.py` or `config_pytorch.py`. The notebook supports both Keras [Chollet et al., 2015] and PyTorch [Paszke et al., 2019] with a common interface for convenience. In more detail, each top-level function is described as follows:

- `get_train_test_from_df(df_train, df_test_A, df_test_B)` extracts, reshapes, strides, and normalises `librosa` features for use as tensors, and saves them to `config.dir_out`, if features with that particular configuration do not exist already. The data is split into train and test based on the matches of experiment ID to the audio tracks from the metadata given in `df_train`, `df_test_A`, `df_test_B`. It is important that no test recordings from these experiments are seen during training in advance, as otherwise model performance is overestimated. Appendix B.3, Table 5 shows the result of feature extraction with baseline feature parameters.

- `train_model(X_train, y_train, X_val=None, Y_val=None)` trains the BNNs on the data supplied (with validation data optional). The assumed input shape is that of the features produced by `get_train_test_from_df()`. The model architecture and training strategies may be changed in `runKeras.py` or `runTorch.py`.

- `get_results(model, X, y, n_samples=1)` evaluates the model object on test data {X, y} with the number of MC dropout samples as `n_samples`. If using deterministic networks, leaving the input argument blank will default to a single evaluation.

## 5.2 Model architecture and training

We extract 128 log-mel spectrogram features with a time window of 30 feature frames and a stride of 5 frames for training. Each frame spans 64 ms, forming a single training example $\mathbf{X}_i \in \mathbb{R}^{128 \times 30}$ with a temporal window of 1.92 s. Test data is strided with the stride length equal to the window size. We list all our parameters affecting the feature transformation in Appendix B.3, Table 4, and include a discussion with general recommendations for feature parameterisation. We supply two benchmark BNN model classes for this dataset:

- **Keras BNN**: A CNN with four convolutional, two max-pooling, and one fully connected layer augmented with dropout layers (shown in Appendix B.4, Figure 3). Its structure is based on prior models that have been successful in assisting domain experts in curating parts of this dataset by thresholding with uncertainty metrics [Kiskin et al., 2021].

- **PyTorch ResNet BNN**: ResNet has achieved state-of-the-art performance in audio tasks [Palanisamy et al., 2020] motivating its use as a baseline model in this paper. We augment the model with dropout layers in the appropriate building blocks to approximate a BNN. We opt to use the pre-trained model for a warm start to the weight approximations. We describe our modifications to the model class in Appendix B.4.

For both models the validation accuracy on a random split of the training data has been used to checkpoint the best-performing model. The code was developed on Ubuntu 20.04 with an i7-8700K CPU, 32 GB RAM and a Titan Xp GPU with 12 GB VRAM, but models were trained and optimised with lower end hardware (Windows 10, Intel i7-4790K CPU with 16 GB RAM and a GTX970 GPU with 4 GB VRAM). We give the number of epochs, the learning rate, dropout rate, the batch size, and discuss ways to further optimise the memory usage in Appendix B.4.

## 5.3 Test results

As a benchmark, we define the test performance with three metrics: the receiver operating characteristic area-under-curve score (ROC AUC), the true positive rate (TPR), also known as the recall, and the true negative rate (TNR), to account for class imbalances in the test sets. These are evaluated over 1.92 second audio chunks. The number of audio samples in each test set following test feature extraction is given in column one of Table 3. Test features are strided by the length of the window to evaluate non-overlapping sections. To simplify the problem, edge cases where the data cannot be partitioned into full 1.92 second sections are removed from the test set. On feature extraction, all

Table 3: Test performance of the four-conv-layer Keras CNN, and two ResNet configurations over the two test sets. The number of 1.92 second samples over which the scores are evaluated is given for mosquitoes by $N_{\mathrm{mozz}}$ and for noise as $N_{\mathrm{noise}}$ respectively. Scores are reported as the mean $\pm$ standard deviation over 10 MC dropout samples.

| Data | Metric | BNN-Keras-4conv | BNN-ResNet-50 | BNN-ResNet-18 |
|---|---|---|---|---|
| Test A | ROC AUC | $\mathbf{0.960 \pm 0.003}$ | $0.959 \pm 0.001$ | $0.918 \pm 0.001$ |
| $N_{\mathrm{mozz}} = 1,714$ | TPR (%) | $71.0 \pm 0.71$ | $\mathbf{95.6 \pm 0.24}$ | $72.64 \pm 0.41$ |
| $N_{\mathrm{noise}} = 2,068$ | TNR (%) | $\mathbf{98.0 \pm 0.25}$ | $73.4 \pm 0.43$ | $90.86 \pm 0.22$ |
| Test B | ROC AUC | $0.349 \pm 0.055$ | $0.545 \pm 0.004$ | $\mathbf{0.670 \pm 0.006}$ |
| $N_{\mathrm{mozz}} = 430$ | TPR (%) | $2.16 \pm 0.48$ | $\mathbf{2.70 \pm 0.50}$ | $1.42 \pm 0.22$ |
| $N_{\mathrm{noise}} = 1,015$ | TNR (%) | $\mathbf{99.8 \pm 0.07}$ | $99.4 \pm 0.25$ | $99.71 \pm 0.03$ |

labels shorter than that window duration are not included in the test set, though this is an area that is left for future work. When comparing performance, we suggest using a test set which has the window size as currently implemented in the code (within `get_feat()` in `feat_util.py`).

Table 3 shows the results that our baselines models were able to achieve. For the intended use case of Test A, all of the models were able to achieve ROC AUC above 0.91. The choice of model to deploy would depend on the preference over error types. For example, ResNet-50 performs better at recalling mosquito events, at the expense of a 26 % false positive rate. On the other hand, the Keras model achieves a false positive rate of only 2 %, but at the expense of missing 29 % of mosquito events. However, performance on Test B is unacceptable by all models, with all of the models categorising nearly all the audio as noise. To verify that the issue does not lie in the test set, after manually verifying each label resulting from feature extraction, we trained the models on half of Test B's recordings, and predicted on the second half, to achieve an ROC AUC of 0.915 (Appendix B.5, Figure 4). Furthermore, prior work was able to achieve ROC AUCs of 0.871 to 0.952 with smaller neural networks which were optimised for use with scarce data [Kiskin et al., 2017]. The task presented in this paper, however, is to be able to achieve good performance over Test B, in addition to Test A, without the model having access to any data (or covariates) from both Test A and Test B.

## 6 Conclusion

In this paper we present a vast database of 20 hours of finely labelled mosquito sounds, and 15 hours of associated non-mosquito control data, constructed from carefully defined recording paradigms. Our recordings capture a diverse mixture of 36 species of mosquitoes from controlled conditions in laboratory cultures, as well as mosquitoes captured in the wild. The dataset is a result of a global co-ordination as part of the HumBug project. The HumBug project is ongoing and the robust recording pipeline described in this paper means that the database will continue to grow in the coming years. A major contribution of this paper has therefore been to link together all the moving parts, from the smartphone sensors and in-house apps, to the curation of a PostgreSQL database with the help of Bayesian neural networks.

Despite decades of work, mosquito-borne diseases are still dangerous and prevalent, with malaria alone contributing to hundreds of thousands of death each year. Therefore a further contribution of this work is to make available mosquito data that is still a scarce commodity. In addition, we have highlighted that our dataset contains real field data collected from smartphones, as well as varying background environments and different experimental settings. As a result, this multi-species data set will continue to help domain-experts in the bio-sciences study the spread of mosquito-carrying diseases, as well as the myriad of factors that affect acoustic flight tone.

Finally, our dataset will be of interest to machine learning researchers working with acoustic data, both in the availability of a real-world acoustic dataset, as well as in the way that we use Bayesian neural networks in the labelling pipeline. We provide simple functions for data manipulation and baseline models in both Keras and PyTorch, alongside extensive documentation. As a result, we make it easy for researchers to start building their own models. It is our aim, by releasing this dataset, to encourage further work in the detection of mosquitoes leading to improved models and better mosquito detection algorithms in the future.

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
