# OpenReview forum: "HumBugDB: a large-scale acoustic mosquito dataset"
_NeurIPS.cc/2021/Track/Datasets_and_Benchmarks/Round1 — Submitted to NeurIPS 2021 Datasets and Benchmarks Track (Round 1)_

### Official Review · Reviewer_mM4X · 2021-06-30
**An excellent submission with new data and benchmark for acoustic mosquito study**

**Rating:** 9
**Confidence:** 3
**Clarity:** yes.

**Strengths:**

1) this paper is very well-written, easy-to-read, and providing very complete information about the code, data, data collection, statistics, ethical concerns, etc.
2) the authors clearly discussed and compared to previous works, and showed that the proposed dataset has many advantages over previous works: a) longer audio, b) better/more accurate labeling.
3) the dataset marks a great effort since it spans 5 years in time and the researchers have collected the data from 5 countries around the globe. Also, the annotation is non-trivial. The use of algorithms to aid annotation is smart and contributing to the annotation techniques for future works.
4) the dataset itself is of good quality, and are demonstrated to be useful for the task of mosquito sound classification, which is of great importance for detecting mosquito and thus preventing diseases transmission.

**Weaknesses:**

I don't have any big concern preventing me from accepting this paper, but below are some minor points if authors want to further improve the work:
1) Can you discuss the data bias? I understand that it's not possible to collect such natural data without any bias. But some discussion about the data bias (e.g. different countries, data imbalance on species), and how would it affect the network training for applications, and how do you practically address these data bias issues, will be very interesting to the readers.
2) Are there other/more applications that your data allows? It would be good to discuss more applications and do more benchmarking on them if possible.

**Additional Feedback:**

no.

Post-rebuttal: I appreciate the authors' great efforts in collecting this data and I think this dataset will be very useful. So I would like to keep my rating.

**Correctness:**

the claims/dataset construction/evaluation methods are correct to my best judgment.

**Documentation:**

yes.

**Ethics:**

no.

**Relation To Prior Work:**

yes.

**Summary And Contributions:**

This paper proposes a better dataset for acoustic mosquito studies, including 20 hours of mosquito audio collected from five countries around the world, with curated labels covering 36 different species and background labels. Using this dataset, the authors also conducted an experiment on the mosquito sound classification task and benchmarked many state-of-the-art algorithms. This paper provides extensive materials for utility code, benchmark code, pretrained models, data release, and additional resources for the data collection process.

---

> ### Author Response · Authors · 2021-07-09
> **(#1) Thank you for your constructive feedback and encouragement**
>
> We thank you for your enthusiasm on the scope of the paper, identifying the effort involved in the collection, curation, maintenance, and presentation of the data. We are particularly pleased that you appreciate the additional material which provides in-depth instructions for code and data use for interested researchers.
>
> ### Weaknesses
> With regards to your minor points for further improvement, we would like to suggest the following:
> 1. Data bias: we believe this is a critical point for training algorithms, so we propose to write a section on data bias for the Datasheet for Datasets and link it with the main text. This would be especially useful for researchers wishing to use the dataset in isolation with just the documentation for the dataset.
> 2. Applications and use cases: we will expand on our data use cases in Appendix D.5 of the Datasheet for Datasets and refer to it from the main text. In particular, we have ongoing work for species classification which utilises this database that will reference this paper in future.
>
> We thank you for your suggestions on improvement, and look forward to incorporating them as soon as possible.

---

> ### Author Response · Authors · 2021-07-10
> **(#2) Update: changes incorporated as a result of your feedback**
>
> To follow up on our list of suggested improvements based on your constructive feedback, we have now completed these in a revised submission (10/07/21). We summarise these changes as follows:
>
> ### Data bias
> We have added an extensive section (D.6) to describe sources of data bias and mitigation strategies for a range of use cases:
> * Nature of mosquitoes: wild and lab grown
> * Recording device
> * Data imbalance
>
> ### Additional applications
> Appendix D.5 has now been re-worked as follows:
> * Past use cases of this dataset added and now referenced in text.
> * Visibility of related work improved with better hyperlinks to data and code repositories.
> * Greatly expanded on the following potential use cases:
>     - **Species classification with HumBugDB.** Training a machine learning model to distinguish between various species.
>     - **Validating species classification models from the literature.**
>     - **Frequency analysis.** Identifying the fundamental and harmonic frequencies of flight tone for a particular species, to improve upon the understanding of bioacoustics literature, and entomological research.
>     - **Examining inter-species (or similar) variability.** For example, the effect on the sound of flight as a result of age, gender, or any field supported in the database.
>
> ### Integration with main text
> To ensure these useful resources do not go unnoticed in the main text, we linked these new sections as follows:
> * Referenced additional use cases (D.5) in the paper structure overview (Introduction), and in the introductory paragraph of Section 5.
> * When discussing statistics of the datasets for training benchmark models (Section 5, paragraph 1), we direct the reader to the new data bias section (D.6).
>
> We would like to thank you once again for these excellent suggestions, and we would be happy to incorporate any additional feedback you may have.

---

### Official Review · Reviewer_XGcR · 2021-07-04
**HumBugDB: a large-scale acoustic mosquito dataset**

**Rating:** 5
**Confidence:** 4
**Correctness:** The benchmark experiment is relativel…
**Clarity:** The paper is well written.

**Strengths:**

+ The collection and labeling of this acoustic mosquito data set are difficult, and it is beneficial for mosquito-borne disease prevention.
+ The paper is well written.

**Weaknesses:**

- The quantification and visualization of statistics and comparisons of datasets are relatively limited.
- I believe such a paper can be well published in a mid-ranked conference but I do not see a sufficient novel contribution to NeurIPS. It is one more vanilla dataset.
- Relevant to researchers in subareas only.

**Additional Feedback:**

See above mentioned.

**Documentation:**

There are some limitations regarding the application of the proposed dataset.

**Ethics:**

Relative comparative advantages and contributions;
Contribution to the community;
More details and statistical information of the data set;
Applications;

**Relation To Prior Work:**

There should be more clear quantification and visualization about the difference from prior work.

**Summary And Contributions:**

This paper collected a large-scale acoustic mosquito data set, including 18 hours of data and 36 categories. The data and baseline code are publicly available.

---

> ### Author Response · Authors · 2021-07-07
> **More clarification needed**
>
> We appreciate your comments on the strengths of our paper. We will now reply to the individual points raised in the review.
>
> ### Weaknesses
> >The quantification and visualization of statistics and comparisons of datasets are relatively limited.
>
> This is a unique dataset, in terms of its size, extent and laborious real-world collection. Comparison with like-for-like data is therefore not easy. What visualization and statistics are you looking for? Please detail your request. We are in the process of adding a table with related datasets (as per Reviewer 1 and 2's request) and will respond once the changes have been made.
>
>
> >I believe such a paper can be well published in a mid-ranked conference but I do not see a sufficient novel contribution to NeurIPS. It is one more vanilla dataset.
>
> Your comment is conjecture, dismissive and at odds with the other reviewers (even those who are constructively critical of the submission). We are aghast with your comment about vanilla datasets. This seems highly dismissive of the years of careful work that has gone into mosquito research and data collection, not just by us, but by many others as well. As to the relevance of the research, we suggest you consider how little data is available to disease (malaria) specialists about the range and activity of mosquitos. Indeed, for some parts of the world, entire policies are based on very small amounts of data. Please also consider that malaria alone is responsible for >400,000 deaths every year in Africa (COVID deaths in Africa are currently [July 2021] around approx 150,000 and no one would call that irrelevant or ‘vanilla’). This dataset is of remarkable size and variation and nothing like it exists. We will certainly make this clearer by including a table of related datasets with useful statistics in the Related Work section (in response to comments made by all reviewers). No prior research has engaged so extensively with field trials in global rural communities to collect data “in the wild” -- previous reliance on lab-cultured mosquitoes does not provide real-world data. The data we and colleagues have collated has the potential to make a difference -- in the way mosquitoes are monitored, and intervention policies are created and assessed.
>
> ### Relation to prior work:
> Please provide more detailed feedback so we can make changes during the review period as appropriate.
>
> ### Documentation:
> What are the limitations you state and how are they relevant to documentation? We provided a very extensive Datasheet for Datasets and have documented all of our code and data to the best of our knowledge and ability.
>
> ### Ethics:
> I believe these points are entered in the incorrect section. Please provide further detail as to how the list of items can be used to improve the paper.

---

### Official Review · Reviewer_Mh2C · 2021-07-06
**Review for HumBugDB: a large-scale acoustic mosquito dataset**

**Rating:** 5
**Confidence:** 5

**Strengths:**

The proposal has several strengths, including:

- The application is socially noble since mosquito-borne diseases affect a large number of people.

- Successful models can have a significant impact on Entomology and directly aid people working in mosquito surveillance.

- The data seems well-organised and curated. It results from a significant amount of non-trivial work by a group of people with different skills.

- The paper is well-written with nice illustrations. There is some space for clarity improvement.

**Weaknesses:**

The main paper weaknesses are:

- There is a tremendous mismatch between the paper motivation and the technical part. The motivation is about mosquito surveillance, including automatic recognition of species. The benchmark is about differentiating mosquito signals from background noise, a much less interesting task.

- The paper lacks a clear explanation of why Test B is so challenging that makes the tested models fail. How is this data partition relevant?

- It seems to be that text is not fair with the relevant literature. The work of Fanioudakis with pseudo-acoustic sensors produce data without background noise and, therefore, would not need to solve the task the released dataset focuses on solving.
Some very relevant work is missing.


**Additional Feedback:**

See my comments above.

**Clarity:**

The clarity of the paper is good. The text is well-written and organised. The figures and tables are clear. The article is well-organised and presented.

The main article flaw is a tremendous mismatch between motivation and reality. The motivation regarding mosquito recognition is highly engaging, but the dataset is about background noise identification. There is nothing in the text discussing data, features or models for species identification.

**Correctness:**

Overall, the dataset and experiments seem to be well-designed and executed. Regarding the data, I do not understand the passage regarding BNN detection on page 4. It seems the positive signals were automatically segmented what may impose a sampling bias on them. It is unclear if many positive passages were left undetected and may appear as negative samples in the dataset.

Also, the text needs to explain better what makes the B partition challenging. The data partition in training and test sets is unclear. It would be much more interesting if the test data would resemble a scenario of application in the real world.

**Documentation:**

It seems to be. I did not check carefully.

**Ethics:**

I do not foresee any ethics issues. Most countries do not require ethical approval for experimental work with live insects.

**Relation To Prior Work:**

The article covers a relevant part of the literature, but it misses a few spots. One example is that the first set of pseudo-acoustic data was released as part of a machine learning competition and is available at the UCR Flying Insect Classification Using Inexpensive Sensors website.

I disagree with the paragraph (85-92) about the work of Fanioudakis et al. Could you explain what "lack of typical background acoustic properties" means? Their work sounds like a better approach for collecting this data since those sensors are insensitive to external factors that generate background noise. Therefore, the problem this paper/benchmark/dataset proposes to solve inexists with data collected with those sensors. Also, the statement about "state-of-the-art" models is very misleading, given the references are for music recognition models, which are naturally designed to work with long samples. However, there are plenty of models that can work with shorter signals.

**Summary And Contributions:**

The paper provides an acoustic dataset of mosquito sounds and a couple of models for distinguishing background noise from signals. The signals were recorded with wild and culture mosquitos for a large number of species. The suggested models perform well in one test set partition and quite poorly in another one. I believe the main idea of releasing the data is to attract researchers and practitioners to test their models in the challenging partition.

---

> ### Author Response · Authors · 2021-07-09
> **Thank you for taking the time for your well-structured and detailed review**
>
> We are glad that you correctly identify the work involved and its potential impact. We politely disagree however about the scope of our paper and differences between acoustic and opto-acoustic paradigms. In the context of acoustic classification, our dataset is a critical resource for improving detection methods, or other use cases of mosquito flight acoustics. We do not see a conflict between the different modalities, they are complementary. We address your criticisms point by point, and summarise the changes we have made (09/07/21) which hopefully address your concerns:
>
> >The main article flaw is a tremendous mismatch between motivation and reality... There is nothing in the text discussing data, features or models for species identification.
>
> Species recognition is important, but can only be performed if a suitable mosquito detector is part of the pipeline. Therefore, our paper motivation is not at all a "tremendous mismatch", but instead a vital component of a real-world predictive pipeline. The database is focused on identifying mosquitoes within the background and on real-world applications of low-cost acoustic sensing in communities in which the disease is prevalent. It is the difficulty of detecting the mosquitoes within noisy backgrounds that motivates us to provide benchmarks for this task. We recognise the importance of species recognition; all species metadata has been preserved to allow researchers to make the best use of this data.
>
> ### Opto-acoustic and acoustic approaches:
> Acoustic detection can be easily deployed at low cost and high-scale due to the integration of low-cost phones. We have demonstrated the ability to use them successfully for such detection tasks. **We have clarified our motivation in the Introduction (lines 47+)**.
>
> ## Relation to Prior Work
> As to a “lack of typical background properties”, the inclusion of background noise addresses the sentiments of Vasconcelos et al., who published their view:
>
> >Finally, none of the published datasets includes environmental noise ..., which is essential to fully characterize mosquitoes in real world scenarios.
>
> For training methods using purely acoustic approaches, a control group of background environmental recording is required to test whether the models are learning the mosquito acoustics or the background conditions.
>
> We can see how the “state-of-art models” comment may come across as misleading.  **We have amended and moved this statement to the acoustic approaches comparison to not discredit any of the approaches utilising opto-acoustic sensors, where those “state-of-art” models would not be useful.**
>
> Furthermore, the opto-acoustic datasets of Chen et al., Fanioudakis et al. use cages of lab raised mosquitoes - these will lack the natural variation found in wild captured specimens [Huho et al., 2007, Hoffmann and Ross, 2018]. This limits the use of their data for the identification of wild populations. **We have included this in the Related Work section**.
>
> ## Minor comments
>
> ### Why Test B is so challenging and how data partitioning is relevant:
> We state reasons for the challenge Test B poses in the first paragraph of Section 5. Furthermore, as the set of data is a test set, it acts as an independent verification to build robust models. We ensured the data is correct, for which we discuss in Section 5.3. The data partition is relevant directly, and critically, as our experiments we supply in Appendix B.5 show that when training on a held-out part of Test B, model performance is more than adequate. However, when the set is disparate to the remaining training data, all of the baselines are unable to achieve good classification performance. Data partitioning is crucial to understanding model performance, which is why we emphasise it so strongly throughout the text. As real-life recordings are often unpredictable, this makes Test B a good candidate as a test-bed for further research.
>
> ## Incorporated changes
> 1. We have strengthened our motivation for using acoustic sensors by adding a paragraph in the Introduction (lines 47+)
> 2. A re-work of the literature review:
>     1. Table (to comply with R1’s suggestions also) to compare notable datasets in the literature, including the work of Chen et al.
>     2. Split discussion into opto-acoustic and acoustic approaches to more clearly explain the context of our contribution and avoid any potential unfair treatment of relevant prior work.
>     3. Made clear the need for longer recording samples for acoustic detection of mosquitoes in free flight with acoustic sensors.
>     4. Added passage on mosquito biodiversity
>
> We thank you for your suggestions to strengthen the quality and impact of our submission. We sincerely hope this answers the points you have raised, and you are satisfied with the changes we have made to the text. We would be glad to incorporate any additional feedback you may have.

---

> > ### Comment · Reviewer_Mh2C · 2021-07-13
> > **re: Thank you for taking the time for your well-structured and detailed review**
> >
> > Thank you for your response. I have some further clarifications:
> >
> > > Therefore, our paper motivation is not at all a "tremendous mismatch" but instead a vital component of a real-world predictive pipeline.
> >
> > It is clear from the text that segmentation is a previous necessary step for species recognition. My complaint is not that the segmentation is irrelevant in the pipeline, but the text virtually ignores the segmentation problem. It comes as a "surprise" in section 4 when it becomes clear that the task addressed in the paper is not species recognition but the separation of signal and background noise.
> >
> > > We recognise the importance of species recognition; all species metadata has been preserved to allow researchers to make the best use of this data.
> >
> > That is great to know. However, as a reviewer, I am assessing this particular paper with its claims and contributions. If the paper "sells" the idea of species recognition, I expect this paper to shows a dataset and models for this end.
> >
> > > Finally, none of the published datasets includes environmental noise ..., which is essential to fully characterize mosquitoes in real-world scenarios.
> >
> > Thanks for this quote. I took a bit of time to check the mentioned paper, and this passage comes with no citation of any source and no supporting experimental evidence. The passage mentions that including wind noise would help to recognise different species. Could you explain how ambient noise such as wind is relevant to species recognition and what evidence you have to support the original claim in your paper? Otherwise, it seems you are reverberating unsupported claims from other papers.
> >
> > > We state reasons for the challenge Test B poses in the first paragraph of Section 5.
> >
> > The explanation in Section 5 is "very difficult low-SNR" and "prominent species not as well represented". Apart from being an imprecise explanation, my question was why it is the case. What makes this data partition particularly interesting for assessment, and why one should invest their time improving performance in this set?
> >
> > Congratulations on your research and your willingness to share data!

---

> > > ### Author Response · Authors · 2021-07-14
> > > **Response to further clarifications**
> > >
> > > Thank you for taking the time to engage in discussion. We offer further clarifications to your comments as follows:
> > >
> > > ### Paper motivation
> > > We have improved the clarity of purpose of our paper - following from the title "HumBugDB: a large-scale acoustic mosquito dataset" which hopefully sets the correct expectations.  Our abstract clarifies the goal of the submission:
> > >
> > > >We present 20 hours of mosquito audio recordings expertly labelled with tags precise in time, of which 18 hours are annotated from 36 different species. Additionally, we provide code to extract features and train Bayesian convolutional neural networks ***that can distinguish mosquito sounds from their corresponding background***.
> > >
> > > We agree it is vital to be clear about the scope of the paper, but we have already reiterated these contributions in the Contributions section:
> > >
> > > > Detailed tutorial code for training state-of-the-art baseline Bayesian neural network models (a range of ResNet and deep CNN models) for the task of ***distinguishing mosquitoes of any species from their background surroundings***, such as other insects, speech, urban, and rural noise.
> > >
> > > The paper summary then points to Section 3 stating:
> > >
> > > >As part of the pipeline, we require a robust method for ***distinguishing mosquito events from  background noise***...
> > >
> > > As a further additional measure, we have re-worded it as follows:
> > >
> > > > ***Due to the rarity of mosquito events, as part of the pipeline we require a robust method for distinguishing mosquito events from background noise. This constitutes the primary use case for the baseline models of Section 5.  We discuss alternate use cases further in Section 5 and Appendix D.5.***
> > >
> > > The data of this paper is primarily focused on the need to improve (acoustic) mosquito detection. This is a prerequisite not just for data acquisition (i.e. 'record-on-detect' functionality) but needed to build a balanced corpus of data prior to any species identification. We have added an extension in a recent revision discussing further use cases of the data, and comment on data bias in Appendices D5, D6 linking this with the main text in Section 5:
> > >
> > > > To showcase the utility of the data, we supply ***baseline models that function as acoustic mosquito event detectors***. Other use cases include, but are not limited to, species classification, harmonic analysis, and the study of inter-species variability. For a more thorough consideration of these use cases refer to Appendix D.5.
> > >
> > > We hope this makes clear the purpose of our work.
> > >
> > > ### Ambient noise
> > > Your statement of:
> > > > The passage mentions that including wind noise would help to recognise different species.
> > >
> > > is misrepresenting the statement made by the original authors (whose work has been peer reviewed and published). To provide better context for their statement, we quote:
> > >
> > > > One crucial research question ... is to what extent the changes induced by environmental conditions (location, temperature, time of day, humidity, air density, etc.) impact the pattern recognition algorithms. The availability of public datasets collected in different geographic and environmental conditions is ***critical*** to understanding how these issues affect recognition algorithms. Finally, none of the published datasets includes environmental noise (e.g. wind or ambient noise), ***which is essential to fully characterize mosquitoes in real world scenarios.***
> > >
> > > We believe the authors are emphasising the need for ***noisy, real-world data*** for the purpose of training, evaluating and deploying robust models. This is exactly what we provide in this dataset.
> > >
> > > By supplying background from the same source, we help build balanced classes for the models, and further provide a control group for scientific investigation. If we use different data sources for background, we introduce additional variables which will lead to confounding when assessing the model's ability to identify mosquito events. An extra utility of this background is that it is sourced from real-world field studies, which is typical of the locations of deployment for this device class. This data is essential for building more robust models for ongoing and future trials.
> > >
> > > ### Challenge of Test B
> > > The dataset shows a failure mode of common models trained on a large quantity of data. It is precisely this negative result that makes this dataset interesting: if we were able to build models in the past that were trained on parts of this data, that perform excellently on the remaining parts, why do models struggle to generalise to this dataset when it is completely withheld? This is an excellent test for robustness: the varying nature of the background and signal, though to our ears as humans does not pose a challenge to labelling, is proving very difficult for the models. This is a research question we motivate with this test set. We hoped to present this as a case example of difficult subsets of test data to inspire improved methods.

---

### Official Review · Reviewer_UKo3 · 2021-07-06
**An interesting dataset containing recordings from different species of mosquitos.**

**Rating:** 4
**Confidence:** 4
**Correctness:** The collection technique and post-pro…

**Strengths:**

1. The dataset can help in recognizing mosquito species from their sounds and thus can help prevent more dangerous species from breeding.

2. The metadata for each audio instance also contains location, time, date (seasons, etc. can be inferred), etc. This can help analyze mosquito patterns and can probably be used beyond machine learning.

3. The data is collected in a wide range of conditions including laboratory settings and in-the-wild settings.

4. The authors use completely unseen test sets which were collected in very different locations under different conditions from what seen by the model during training. The evaluations of future works can potentially be very stable and point towards robustness.

5. The mobile app and database schema is very clearly explained. The codebase and the dataset are public.

**Weaknesses:**

1. The authors should describe the modifications they made to a bed net that attracted mosquitos to fly close to the smartphone. The authors should also describe mosquito behavior that they exploited instead of just citing previous research work. I did not find the previous work online and was not able to verify this step.

2. Multiple parts of the data are actually recorded using a Telinga. Some description about this device and its usage for this project will be necessary for Section 3.

3. The baseline models should be renamed appropriately. More baselines like VGG16 and Inception Net should probably be tried. I also believe, finetuning (transfer learning) on VGG16 trained on other audio-related tasks will be interesting and can improve performance.

4. The performance of the models also varies widely in the two test sets. Detailed experiments should be done to analyze this. Furthermore, ResNet-18 and ResNet-50 are essentially similar architectures, with ResNet-50 being deeper. However, the trend of both of these models reverses in Set B when compared to Set A for ROC-AOC. Why is this happening?

5. In line 324, the authors mention prior works which were not mentioned in the table. In my view, even they should be mentioned in the table, and their performance should be reported on this dataset.

6. The species distribution is not clear in the graphs given in Figure 6 (supplementary material). Also, it will be really beneficial to have some information about the disease caused by each of these species. By reading the paper, I felt that only malaria-causing species were considered while opportunities existed to focus on other diseases like Dengue.

7. No clear policies and strategies for dataset maintenance and update are mentioned in the paper. The dataset can and should be improved over time with more species, more data per species, etc.

Overall this dataset suffers from baseline evaluations which are necessary to understand the usability of the dataset. I would ideally like to see more rigorous evaluations and more experiments to focus on how the dataset can be used.


**Additional Feedback:**

1. Please put a small table in the related works comparing each dataset in terms of hours, expertize in labelling, background noise present or not, etc. It is very helpful for the reader to get the summary along with the long text in the related works section.

2. I did not understand the usage of Keras BNN and PyTorch ResNet BNN. Both the libraries support similar applications and could have been implemented using a common library.

**Clarity:**

The paper is not too well written. A table to compare past datasets with the proposed one will be a great addition. The graphs in supplementary are not easily readable. Overall the writing quality should be improved to make the contributions clearer.

**Documentation:**

The dataset is publicly available and collection steps are very clearly stated. The database system used,  mobile application etc, is very nicely explained.

**Ethics:**

The authors clearly mention the ethical concern of having background human speech. They have taken specific permissions for each instance and I have no ethical concerns.

**Relation To Prior Work:**

Prior work is stated well.

**Summary And Contributions:**

The authors present a dataset containing about 20 hrs of acoustic mosquito recordings, out of which 18 hrs are labeled by experts (involving 36 different species). The dataset contains recordings from various locations, including those bred indoors in culture cages and those captured in the wild. The authors have also provided a public codebase that can be used to extract acoustic features and Bayesian NN-based models to separate the mosquito sounds from the background. The dataset can be used by machine learning experts, entomologists, etc., to model mosquito behavior and prevent outbreaks of various mosquito-borne diseases like Malaria.

---

> ### Author Response · Authors · 2021-07-09
> **Author response to review and updated submission based on your feedback**
>
> Thank you for taking the time to review our work and offer us suggestions. We have already taken steps to improve the paper based on your feedback, and have updated our submission (09/07/21). A few of your criticisms are actually direct results of Strength 4 which you identified, that we carefully ensured not to violate. Furthermore, some points are addressable as a simple oversight (e.g. (1), (7)). Our response to each individual point is as follows:
>
> ### Weaknesses
> 1. In lines 127 and 179 of the original submission we have cited Sinka et al., 2021 which answers this question and offers citations for mosquito behaviour:
>  >The bednet is adapted by the addition of a second outer canopy and a detachable pocket. The pocket is placed at the highest point of the outer canopy above the occupant’s head and holds a budget smartphone running the MozzWear App. The occupant switches on the App as they enter the bednet at night. Host-seeking mosquitoes are attracted to the CO2 in the breath of the occupant and become trapped within the second canopy of the HumBug Net. Here they naturally migrate to the highest point of the net (a behaviour common amongst flying insects and exploited in insect trapping methods such as the Malaise Trap (Mississippi Entomological Museum; Evans 2016) ...
>
>     **To make this information more easily accessible we added “Section 2.1.2” ([Sinka et al. 2021, Sec. 2.1.2]) to any citation  made with reference to the adapted bednet.**
>
> 2. Section 3 describes the deployment pipeline, in which smartphones are the target. The Telinga device, however, was used to collect previous data (described in Section 4). **We added more detail on recording devices for the Datasheet for Datasets Appendix D.3 and cite this section in the paper upon first mention of the Telinga device (line 258)**
>
>
> 3. The ResNet50 application is already a transfer learning paradigm (described in Appendix B.4. PyTorch ResNet-X), and is commonly encountered in literature. As this is a dataset paper, we see the baselines as sufficient (having performed well in the past [Kiskin et al., 2019, 2020] and also in similar applications in literature [Palanisamy et al., 2020]). Maximising classification performance on this dataset is outside of the scope of this track.
>
> 4. The TPR of all three models is <3% on Test B, indicating the failure mode for all the models is consistent. The behaviour of all the models is also consistent across Test A. We have conducted further experiments with Test B to ensure data integrity and included these in B.5, which we explain in Section 5.3. **We note that our reference originally pointed to B.4, which was a typo we corrected to B.5**. The purpose of the inclusion of Test B as a true out of sample test, is that it represents an unpredictably difficult scenario that the models are likely to encounter in deployment.
>
> 5. We cannot include that model in the same table as it has been designed and trained on one part of Test B and evaluated on another part. In this paper, however, Test B is held out completely from training (as you identify as Strength 4). The accuracy figures of the cited prior work are only a like-for-like comparison with Appendix B.5, where similar performance was observed, even though the prior model was tuned to that dataset.
>
> 6. **On the clarity of Figure 6, we added Table 6 showing the exact figures of species distribution per recording location and modality.** In line 188 of the original submission (Section 4.2.1) we specifically list the vectors of each disease:  “These include primary malaria vectors (e.g. Anopheles gambiae, An. arabiensis), arbovirus vectors including primary vectors of dengue virus (Aedes albopictus), yellow fever virus (Aedes aegypti) and west nile virus (Culex quinquefasciatus).”
>
> 7. Database maintenance in accordance with NeurIPS guidance on Datasheets for Datasets is described in Appendix D.6. We used a repository which resolves to the latest version with one DOI, and GitHub commits will be made for updated metadata and code. New data with species information is expected soon (recorded in the DRC), though due to COVID-19 no exact timelines can be supplied.
>
> ### Additional feedback and other comments
> 1. This is an excellent suggestion. **To clarify our contribution relative to related works, we re-wrote Related Work and added Table 1 which lists the data source, the number of samples, the sample duration, the sensor, the number of species, and the type of mosquito surveyed.**
> 2. The strong point of the code framework is that both libraries are natively supported with a common interface. This leaves practitioners free to choose whichever framework they are most comfortable with. We would be happy to port the keras model to PyTorch if it improves uptake in future.
> 3. **We have reiterated our contributions in the Contributions section, and added our motivation for using acoustic devices in the Introduction (lines 47+)**

---

### Author Response · Authors · 2021-07-10
**Revision changelog**

To simplify keeping track of all edits made to the submission during the author response period, we provide a concise summary for the convenience of reviewers and meta reviewers:

(09/07/21 - 10/07/21)
* Added Table 6 with species distribution per recording location to improve clarity of Figure 6
* Corrected reference to Appendix B.5 which discusses Test Set B further
* Improved reference to [Sinka et al. 2021, Sec. 2.1.2] explaining bednets and mosquito behaviour
* Added detail on recording devices in Appendix D.3 and linked in text
* Re-worded contributions in Contributions section
* Added motivation for using acoustic devices in Introduction
* Added sentence to motivate use case for detecting mosquito events as part of a pipeline in Section 3.1.
* Re-worked literature review:
    - Table to compare notable datasets in the literature, including the work of Chen et al.
    - Split discussion into opto-acoustic and acoustic approaches to more clearly explain the context of our contribution and avoid any potential unfair treatment of relevant prior work.
    - Amended and moved “state-of-art” comment to acoustic approaches comparison to not discredit any of the approaches utilising opto-acoustic sensors.
    - Made clear the need for longer recording samples for acoustic detection of mosquitoes in free flight with acoustic sensors.
    - Added passage on mosquito biodiversity
* Added section on data bias (D.6):
    - Nature of mosquitoes: wild and lab grown
    - Recording device
    - Data imbalance
* Appendix D.5 has now been re-worked as follows:
    - Past use cases of this dataset added and now referenced in text
    - Visibility of related work improved with better hyperlinks to data and code repositories
    - Greatly expanded on four potential use cases
* Referenced use cases in the paper structure overview (Introduction), and in the introductory paragraph of Section 5.
* When discussing statistics of the datasets for training benchmark models (Section 5 paragraph 1), we direct the reader to the new data bias section (D.6).

(14/07/21)
* Re-worded section on intended use cases of data in Section 3.1 to better link with Section 5 and D.5 to bulletproof the contributions and scope of our paper.

---

### Decision · Program_Chairs · 2021-07-26

**Decision:**

Reject

**Comment:**

Although the paper proposed a dataset for an extremely important and relevant issue, reviewers felt that the final task the authors wanted to solve was not evaluated. There is a gap between the motivation (insect surveillance) and the task that is evaluated using the dataset (background removal).